# Seeing an apocalyptic post-antibiotic future lowers antibiotics expectations and requests
Miroslav Sirota ✉ & Marie Juanchich

## Abstract

**Background** Antibiotic resistance is an ongoing pandemic which represents a global public health threat. To encourage the judicious use of antibiotics, public health discourse and campaigns often engage in threat-based messaging depicting an apocalyptic post-antibiotic future. We studied the effectiveness of the strategy because of mixed evidence for its success, and because it is unclear how experiencing the COVID-19 pandemic might have influenced it.

**Methods** We conducted a randomised controlled trial with 378 participants in three waves (before and during the pandemic in 2021 and 2022). Participants were randomly allocated to either the baseline arm, featuring a control film, or the intervention arm featuring a short film, Catch, depicting a post-antibiotic future. Participants expressed expectations and intended requests for antibiotics for a hypothetical ear infection and their adherence to a prescribed antibiotic for a hypothetical kidney infection. In waves 2 and 3, they also reported any COVID-19-related changes to their antibiotic desires.

**Results** Showing participants a film about a post-antibiotic future substantially lowers clinically inappropriate expectations for antibiotics and their intended requests. Participants report that the experience of the COVID-19 pandemic decreased their desire for antibiotics but only when they watched the intervention film. The intervention slightly decreases participants' adherence intentions towards a prescribed antibiotic treatment.

**Conclusions** Presenting a film about an apocalyptic post-antibiotic future lowers expectations and intended requests for antibiotics and therefore has the potential to encourage judicious use of them. However, the adverse effects of such messaging on adherence to a course of antibiotics should be proactively managed.

## Plain language summary

When bacteria evolve to resist the effects of an antibiotic, often due to repeated exposure, it leads to drug-resistant infections. This antibiotic resistance puts modern medicine at risk as it renders these infections increasingly difficult to treat with standard antibiotics. To avoid unnecessary antibiotic use, public health campaigns sometimes use threat-based messaging about an apocalyptic future in which antibiotics do not work at all. However, it is not clear whether these messages work as intended. In our study, we found that showing people a future where antibiotics do not work made them less likely to want and ask for unnecessary antibiotics for a hypothetical self-limiting infection. People also reported that their experience of the COVID-19 pandemic decreased their desire for antibiotics. However, the apocalypse messaging also made people slightly less likely to take prescribed antibiotics. Overall, showing people a glimpse into a post-antibiotic future may encourage more careful antibiotic use.

Recently, people around the globe have been experiencing the COVID-19 pandemic. However, another global pandemic is unfolding in front of our eyes[1]. Antimicrobial resistance caused 1.27 million deaths and played a role in 4.95 million deaths in 2019 across the world[2]. Even though the evolution of resistance is a biological process, human behaviour such as the unnecessary consumption of antimicrobials in agricultural production and human medicine substantially speeds up this process[3]. Therefore, enabling the public to understand how they can change their behaviour—for instance, using antibiotics judiciously and appropriately—could help us to curb antimicrobial resistance.

One important and changeable set of behaviours contributing to antibiotic resistance concerns the public's consumption of antibiotics for self-limiting infections for which antibiotics are not needed[4,5]. In countries where there are few regulations imposed on the prescribing of antibiotics, the public can either buy them directly over the counter or on the Internet[6]. In countries with strict regulations, patients can still implicitly or explicitly obtain prescribed antibiotics from health professionals. Indeed, recent evidence shows that patients' expectations and requests for antibiotics substantially contribute to over-prescribing and, in turn, the overconsumption of antibiotics[7–10]. For instance, according to one experimental vignette study, family physicians were twice as likely to prescribe antibiotics to a patient asking the physicians to do something about their illness compared with the control group where this was not mentioned. In the manuscript presented here, we investigate whether communicating about antibiotic resistance

Department of Psychology, University of Essex, Wivenhoe Park, CO4 3SQ Colchester, UK. ✉e-mail: msirota@essex.ac.uk

post-apocalyptic future can reduce clinically inappropriate antibiotic expectations and intentions to request antibiotics.

## The effects of a post-antibiotic future

Public health organisations and authorities routinely engage in various communication strategies and media discourses that vividly portray the risks associated with antibiotic resistance. Often, these depict the worst-case scenario of a post-antibiotic era in which antibiotics do not work[11–14]. The idea behind this strategy is that people will become more aware of the potential consequences of their overuse of antibiotics, which can then lead to greater efforts to prevent antibiotic overuse. For example, during the 2017 World Antibiotic Awareness Week campaign organised by the World Health Organization, a short film entitled Catch was released. It tells the story of a father whose daughter is fighting a bacterial infection for which antibiotics do not work anymore. This same infection has led to the death of her mother and younger brother and the father fails to prevent his daughter from contracting the illness. The film uses vibrant melancholic piano music to introduce a man wearing a hazmat suit bringing a food tray to the man's daughter who is locked in her bedroom. The father and daughter live as recluses and avoid any contact, being scared of touching any items. The film is intense and shows a world where any exposure to a bacterial pathogen would be deadly, where a cough means having to be locked down and possibly die.

Depicting a post-antibiotic apocalyptic future became a common communication strategy not only in public campaigns but also in the public engagement discourse of medical professionals. For instance, in her book The Drugs Don't Work, Professor Dame Sally C. Davies, the former Chief Medical Officer for England, describes a story of a near post-antibiotic future. She tells the experience of a woman called Mrs Xu who is isolated from her loved ones in her home, dying from bacterial illnesses for which antibiotics do not work anymore[15].

Such messaging might be an effective way to raise awareness of the problem of antibiotic resistance and change people's behaviour by decreasing their expectations and use of clinically inappropriate antibiotics. However, evidence on the effectiveness of such campaigns is limited and mixed, as they often lack control groups and involve complex interventions with multiple components[16].

On one hand, there is theoretical and empirical support for the prediction that, in general, fear-based messaging based on depiction of a threat can be effective. Decision-making theories suggested and empirically demonstrated that perceived risks weigh more heavily than benefits in the evaluation of options[17]. Therefore, highlighting those might be more effective in influencing people's behaviour. For instance, when the utility-based signal detection theory is applied to antibiotics expectations, increasing the risk of antibiotic resistance decreases the desirability of antibiotics, especially in situations when the decision-maker faces the uncertainty of whether or not they are needed[18]. Aligned with such a prediction, experimental evidence shows that messages making the costs of antibiotic resistance salient decrease inappropriate antibiotics expectations[18,19]. Although their effect is not always consistent, fear-based messages were shown to be effective in changing attitudes, intentions and behaviours across several health-related domains in a recent meta-analysis of 127 research articles, especially if those messages were coupled with clear action to prevent the threat[20].

On the other hand, some evidence suggests that apocalyptic messaging may not be as effective as hoped. For example, the influential report mandated by the Wellcome Trust does not recommend messaging about the antibiotic apocalypse since it is not deemed to be credible[13]. However, this recommendation was based on the perception of the low plausibility of such a scenario reported by the members of the public in a focus group, which was conducted before the COVID-19 pandemic. Furthermore, prior research in other health-related contexts has documented that fear-based messaging can lead to negative effects, trigger negative emotions, and even backfire, which can ultimately lead to individuals disregarding the message altogether[21]. Moreover, such messaging may have adverse effects rarely investigated in public health messaging such as lower adherence to prescribed antibiotics when these are clinically justified[16].

Finally, from the theoretical point of view, even if threat-based messaging were effective it is not clear what the optimal level of elicited fear is. According to the linear model of fear theory, higher levels of fear would result in greater motivation to adopt the recommended behaviour[22]. In contrast, the curvilinear model of fear theory predicts that high levels of fear would lead to defensive avoidance and make the message less effective[23]. In the last up-to-date meta-analysis, the linear model was supported, as higher levels of fear were more effective than moderate levels in changing attitudes and behaviours[20].

In summary, while apocalyptic messaging is commonly used to address antibiotic resistance, there is mixed evidence of its effectiveness. High-level fear messages are effective but they may be seen, at least for some people, as implausible. In light of the COVID-19 pandemic, perceptions of the potentially apocalyptic consequences of infectious diseases may have shifted. Consequently, apocalyptic messaging may be seen in the light of this experience to be more credible, which might reduce its backfiring effect. Further research is needed to better understand this kind of messaging to communicate the facts about antibiotic resistance and change behaviour.

## Present research

In the research presented here, we investigate the effect that watching a film about a post-antibiotic era has on people's clinically inappropriate expectations and requests for antibiotics and their intention to adhere to a prescribed course of antibiotics. We collect data from three different samples at three different times: before the COVID-19 pandemic (wave 1), during the COVID-19 pandemic during the last lockdown phase (wave 2), and in the post-lockdown phase when restrictions are lifted (wave 3). We adopt an intervention created to raise awareness of the risk of antibiotic resistance—the WHO short film Catch depicts a post-antibiotic era in which antibiotics do not work.

We hypothesise that the intervention (vs. baseline) decreases the inappropriate expectations for antibiotics (hypothesis 1) and requests for antibiotics (hypothesis 2). We also hypothesise that, with the experience of the COVID-19 pandemic, the self-reported desire for antibiotics decreases in the intervention condition compared with the midpoint and the baseline group (hypotheses 3 A and 3B, respectively). Finally, we hypothesise that the intervention does not diminish adherence to a prescribed course of antibiotics in a clinically appropriate situation (hypothesis 4).

We find that the intervention, compared to the baseline, decreases the inappropriate expectations for and requests for antibiotics. Participants report that their experience of the COVID-19 pandemic reduces their desire for antibiotics, but this effect is observed only after watching the intervention film. Finally, the intervention also slightly diminishes adherence to a prescribed course of antibiotics in a clinically appropriate situation.

## Methods
### Participants

We collected data from the different groups of UK participants, each at a different point in time (i.e., waves). Wave 1 was the Pre-COVID wave, from 13/2/2018 to 16/03/2018, before the pandemic. Wave 2 was the COVID lockdown wave from 24/2/2021 to 28/5/2021, during and shortly after the third and last lockdown which ended on 29 March in the UK. Wave 3 was the COVID post-lockdown wave from 11/11/2021 to 02/5/2022, when restrictions were gradually lifted despite the spread of the Omicron variant. The dates represent enrolment of the first and last participant in each wave. The participants were members of the public and students recruited locally. We determined our stopping rule a-priori (i.e., to reach the target sample size in each wave or finish by a certain date). We aimed to recruit 128 participants in each wave based on an a-priori power analysis to be able to detect a medium effect size (Cohen's $d = 0.5$) assuming $\alpha = 0.05$ and $1 - \beta = 0.80$ for an independent-samples $t$-test. In the Pre-COVID wave, a sample of 118 participants was collected; the data collection ceased by a pre-specified date even though the sample size was not reached. In waves 2 and 3,

the desired sample size that assumed a small attrition rate was reached ($n_2 = 136$, $n_3 = 135$, respectively). Thus, the total sample size was $N = 389$. Following the a-priori exclusion criterion, 11 participants were excluded because they failed to watch the video fully (i.e., less than 825 s in the control condition and less than 910 s in the intervention condition).

The analytical sample consisted of 378 participants (ages ranging from 18 to 59 years, $M = 22.6$, $SD = 7.0$ years; 73.0% of whom identified as female, 26.5% as men, and 0.5% selected another option). The levels of the participants' education were relatively heterogeneous: 0.3% did not complete their high school education, 59.9% completed high school education, 36.8% completed a college degree, 3.7% completed a master's degree and 0.3% completed a PhD or other professional degree. In a post-hoc sensitivity analysis, we determined that the achieved analytical sample size allowed us to detect at least a small effect size of Cohen's $d = 0.29$ in the total sample assuming the same parameters as above and at least a medium effect size in each wave ($d = 0.55$, $d = 0.48$, and $d = 0.49$). Participants were recruited in laboratory conditions in wave 1 and online in waves 2 and 3. Participants were eligible to take part if they were at least 18 years old. Their participation in a 30-minute study was voluntary and, in the case of students, rewarded with course credits.

### Design

In a between-subjects experiment, participants were randomly allocated either to a baseline condition ($n = 192$), where they watched a science-fiction film that did not focus on antibiotics, or the Catch video clip intervention depicting a post-antibiotic era ($n = 186$). The participants then assessed their expectations and intentions to request antibiotics for a hypothetical ear infection and their adherence to a course of prescribed antibiotics for a bacterial kidney infection. The study was a single-blind randomised controlled trial—the random allocation of the participants was done by the Qualtrics built-in randomiser operating automatically using the Mersenne Twister algorithm[24].

### Materials and procedure

After giving informed consent, the participants watched one of the two short films based on their random allocation. In the baseline condition, they watched the film Digital Antiquities (itvs.org/films/digital-antiquities), which is situated in the near future (2036). The film tells the story of a woman who is working on recovering data from old devices coming from a pre-cloud era. She recovers data from a CD for a customer and finds out that he is her brother. In the intervention condition, the participants watched the film Catch (www.catchshortfilm.com/) described in the Introduction. A total of 17 participants reported that they had seen the film before but this did not affect their responses (see Results). The films were matched for a similar length (15′ 13″ vs. 16′ 00″ minutes) and both feature realistic scenarios based on our projections of the current state of the world in the near future. We collected the data from wave 1 in the lab and from waves 2 and 3 remotely using the same online questionnaire. Since the data collections in waves 2 and 3 were conducted remotely online, we used audio and video check tasks before randomising the participants into the conditions. Only the participants who successfully passed these checks were allocated to the conditions. For the sound and video checks, participants were asked to identify a letter from a visually presented $3 \times 5$ matrix of letters according to matrix coordinates spoken by a researcher recorded in a short video.

After watching the film, the participants read a vignette about a viral ear infection and expressed their expectations for antibiotics and their intentions to request them. In the ear infection vignette, the participants imagined that they had an ear infection which presented with fever and pain and that they went to consult their family physician. The participants learnt from the physician that the ear infection was likely of viral aetiology and would clear up by itself. The viral ear infection vignette was designed to assess inappropriate antibiotic expectations and intentions to request antibiotics. The description of the symptoms and the clinical judgement with no antibiotic prescribing followed the National Institute for Health and Care Excellence (NICE) guidelines for prescribing antibiotics for ear infections and were

developed in consultation with a family physician in prior research[10]. Participants indicated their expectations for antibiotics by (dis-)agreeing with four items presented in a random order. These were: "I should get a prescription for antibiotics", "I should be offered a prescription for antibiotics", "I would want my doctor to give me a prescription for antibiotics" and a reversed item "I would not want my doctor to offer me a prescription for antibiotics". Their reactions were recorded on a 6-point Likert scale (1 = Strongly Disagree, 2 = Disagree, 3 = Mildly Disagree, 4 = Mildly Agree, 5 = Agree, 6 = Strongly Agree). The internal consistency of the scale was excellent (Cronbach's α = 0.91, 95% CI [0.89, 0.92]), so we used an average score. The participants indicated their intentions to request antibiotics by answering four questions presented in a random order. These were: "I would request a prescription for antibiotics", "I would mention antibiotics to my doctor", "I would suggest that I should have antibiotics" and "I would demand a prescription for antibiotics". These were rated via a 6-point Likert scale (1 = I certainly would not, 2 = I would not, 3 = I probably would not, 4 = I probably would, 5 = I would, 6 = I certainly would). The internal consistency of the scale was excellent (Cronbach's α = 0.90, 95% CI [0.89, 0.92]), so an average score for each participant was used for analysis. Finally, participants in waves 2 and 3 (i.e., during the COVID pandemic), reported their change in desire to have antibiotics while experiencing the COVID-19 pandemic. ("How has the experience of the COVID-19 pandemic changed your desire to get antibiotics for the symptoms described in the previous scenario?") They did so by completing a sentence ("Now, I would wish to get antibiotics… than before the COVID-19 pandemic") by choosing one of the qualifiers from a 7-point Likert scale (1 = Much less, 2 = Less, 3 = Slightly less, 4 = About the same, 5 = Slightly more, 6 = More, 7 = Much more).

The participants also read a vignette about a bacterial kidney infection. They imagined that they had the symptoms of a kidney infection: fever, pain, a burning sensation during urinating, and blood in their urine, and that they consulted their family physician. The participants learnt from the physician that the kidney infection was caused by bacteria and that a course of antibiotics was needed to treat it following the NICE guidelines[25–27]. The participants were asked whether they would adhere to a prescribed course of antibiotics (i.e., "Would you take the 14-day course of prescribed antibiotics as recommended by your GP?") using a dichotomous scale (No = 0, Yes = 1), with options presented to them in a random order.

The participants then answered four follow-up questions. To check the effect of the manipulation, the participants reported whether the film influenced their antibiotics expectations ("To what extent has the short film you have just watched affected your expectations for antibiotics?") This was rated on a 5-point Likert scale (anchored as 1 = Not at all to 5 = Very much). Those who claimed that they were affected at least minimally (i.e., selected options 2 to 5) were then asked to describe what specifically affected their expectations for antibiotics. ("Describe briefly what aspect of the short film you have watched affected your expectations for antibiotics.") To control for prior knowledge of the film, we asked whether participants had heard about World Antibiotic Awareness Week in 2017 when the film was released and also about the film itself (see Supplementary Methods).

### Transparency and openness

We conducted the study in accordance with the ethical standards of the American Psychological Association (APA) and obtained ethical approval from the Ethics Committee of the Department of Psychology at the University of Essex. We have reported how we determined our sample size, all measures, manipulations and exclusions. The study was partially pre-registered: we did not pre-register it before launching the data collection in wave 1 but we pre-registered our hypotheses, measures and analyses before collecting data in wave 2 (https://aspredicted.org/am6hp.pdf) and before collecting the data in wave 3 (https://aspredicted.org/ie7x4.pdf). The data set, codebook, R code, pre-registrations and materials are available at https://osf.io/rmvck/[28,29].

### Statistics and reproducibility

We conducted a series of independent sample t-tests to assess the effect of manipulation on antibiotics expectations and intended requests, both

overall and within each wave. We also performed a series of one-way analyses of variance to examine the main and interaction effects of the manipulation and wave on antibiotics expectations and intended requests. Additionally, we also performed a series of one-way analyses of covariance to examine the same effects while controlling for socio-demographic variables. We performed an independent sample $t$-test to assess the effect of manipulation on self-reported changes in antibiotic desire due to COVID-19, and a one-sample $t$-test to assess the self-reported changes in each condition. Finally, we used a chi-squared test to assess the effect of manipulation on antibiotic adherence.

We supplemented all hypotheses tests by computing a Bayes factor with default priors using the BayesFactor package in R[30]. Broadly, a Bayes factor quantifies the ratio of the likelihood of observing the data under model A (for instance, a model positing an effect of the intervention) versus that under model B (for example, a model positing no effect).

All statistical analyses were conducted using the R programming language[31] (v 4.1.1).

### Reporting summary
Further information on research design is available in the Nature Portfolio Reporting Summary linked to this article.

## Results
### Manipulation check
Participants judged that their expectations for antibiotics were more strongly affected by the film featured in the intervention condition ($M = 2.7$, $SD = 1.3$) than the film in the baseline condition ($M = 1.3$, $SD = 0.7$), $t(376) = -13.56$, $p = 2.31 \times 10^{-34}$, Cohen's $d = -1.40$. Only 23.7% of participants in the intervention condition claimed not to be affected at all compared with 82.6% of participants in the control condition. Those affected at least to some extent described the aspects of the film affecting their expectations for antibiotics in their open-ended responses.

In the intervention condition, where participants watched the "Catch" film, the main factual theme in these responses was the impact of antibiotic resistance on personal and societal health, highlighting the overuse of antibiotics and the importance of using them only when necessary. Some viewers also mentioned the film's impact on their understanding of the long-term effects of antibiotics and the risk of antibiotic resistance in the future. Finally, the participants who completed the study post-COVID onset (waves 2 and 3) also voiced their concerns about the proximity of the reality portrayed in the film to the reality of the COVID-19 pandemic. The responses from the minority of participants in the baseline condition, who said that their expectations were affected by the film in the previous question, primarily conveyed topics relating to the general themes of waste and overuse, and the potential negative consequences of such overuse, such as antibiotic resistance. Thus, we considered our intervention to be an effective manipulation of the harms associated with antibiotic resistance.

### The intervention effect on antibiotics expectations
Watching the post-antibiotic apocalyptic film substantially decreased the inappropriate expectations for antibiotics compared with the baseline group, $t(376) = 6.10$, $p = 2.68 \times 10^{-9}$, Cohen's $d = 0.63$ (Fig. 1, panel B). A Bayes factor analysis yielded decisive evidence to support the intervention effect model relative to the null effect model, $BF_{10} = 34.3 \times 10^5$. The total effect remained significant even after controlling for prior knowledge about the film and the 2017 World Antibiotic Awareness Week, $F(1, 373) = 37.99$, $p = 1.84 \times 10^{-9}$, Cohen's $f = 0.32$. Furthermore, the effects were significant within each of the three data collection waves (Fig. 1, panel A): $t_{wave1}(105) = 3.81$, $p = 2.36 \times 10^{-4}$, Cohen's $d = 0.74$, $BF_{10} = 10.3 \times 10^1$; $t_{wave2}(134) = 4.73$, $p = 5.69 \times 10^{-6}$, Cohen's $d = 0.81$, $BF_{10} = 28.6 \times 10^2$; $t_{wave3}(133) = 2.37$, $p = 0.019$, Cohen's $d = 0.41$, $BF_{10} = 2.3$; respectively. Despite some variation of the effects across the waves, the wave and the effect magnitude did not interact, $F(2, 372) = 1.19$, $p = .304$, while the main effects of the intervention and wave were significant, $F(1, 372) = 38.44$, $p = 1.50 \times 10^{-9}$; $F(2, 372) = 5.99$, $p = 2.75 \times 10^{-3}$, respectively. So, as we can

see in Fig. 1, the expectations for antibiotics were generally higher in waves 2 and 3 than in wave 1. However, this could be due to sampling variability in our samples. To decrease the likelihood of such a possibility, we re-ran the ANOVA analysis while controlling for socio-demographic variables of age, gender, and education level. In this subsequent exploratory analysis, the interaction term remained nonsignificant: $F(2, 365) = 1.68$, $p = 0.187$, while the effect of the intervention and wave continued to be statistically significant, $F(1, 365) = 38.39$, $p = 1.56 \times 10^{-9}$; $F(2, 365) = 4.72$, $p = 9.46 \times 10^{-3}$, respectively. Thus, the variability of the samples in the measured socio-demographic variables could not account for the observed effects.

### The intervention effect on intentions to request antibiotics
Similarly, the cinematic depiction of a dramatic post-antibiotic future lowered the intentions to request antibiotics compared with the baseline (see Fig. 1, Panels C and D). This was the case for the overall effect, $t(376) = 5.09$, $p = 5.77 \times 10^{-7}$, Cohen's $d = 0.52$, $BF_{10} = 21.1 \times 10^3$. (The effect remained significant even after controlling for prior knowledge about the film and World Antibiotic Awareness Week 2017, $F(1, 373) = 26.79$, $p = 3.72 \times 10^{-7}$, Cohen's $f = 0.27$.) And it was the case for the partial effects in each of the three waves: $t_1(105) = 2.30$, $p = 3.39 \times 10^{-3}$, Cohen's $d = 0.58$, $BF_{10} = 10.3$; $t_2(134) = 3.18$, $p = 1.81 \times 10^{-3}$, Cohen's $d = 0.55$, $BF_{10} = 16.8$; $t_3(133) = 2.83$, $p = 5.41 \times 10^{-3}$, Cohen's $d = 0.49$, $BF_{10} = 6.6$; respectively. Despite some variation of the partial effects across the waves, the wave and the effect magnitude did not interact, $F(2, 372) = 0.03$, $p = 0.972$. The main effects of the intervention and wave were significant, $F(1, 372) = 26.09$, $p = 5.22 \times 10^{-7}$; $F(1, 372) = 4.86$, $p = 8.26 \times 10^{-3}$, respectively. The requests for antibiotics were slightly higher in waves 2 and 3 than in wave 1, but again this might be due to sampling variability. Consequently, we rerun the same analysis, while controlling for the sociodemographic variables of age, gender, and education. The interaction term remained nonsignificant, $F(2, 365) = 0.12$, $p = 0.891$, while the effects of the intervention and wave remained to be statistically significant, $F(1, 365) = 25.19$, $p = 8.13 \times 10^{-7}$; $F(2, 365) = 4.73$, $p = 9.39 \times 10^{-3}$, respectively. Thus, similarly to antibiotic expectations, the variability of the samples in the measured socio-demographic variables could not account for the observed effects in the intentions to request antibiotics.

### The intervention effect on the perceived changes in desire due to the COVID pandemic
Overall, the participants reported that their desire for antibiotics was slightly lowered due to the COVID-19 pandemic experience ($M = 3.8$, $SD = 1.1$; the middle point 4 indicating no change), $t(270) = -3.81$, $p = 1.75 \times 10^{-4}$, Cohen's $d = -0.23$ (tested against the middle point of 4). When this reduction was unpacked per condition, the decrease was statistically significant in the intervention condition ($M = 3.6$, $SD = 1.3$), $t(133) = -3.59$, $p = 4.58 \times 10^{-4}$, Cohen's $d = -0.31$, $BF_{10} = 40.4$ but not in the baseline condition ($M = 3.9$, $SD = 0.9$), $t(136) = -1.58$, $p = 0.116$, Cohen's $d = -0.13$, $BF_{01} = 3.1$. This difference between the control and intervention was also statistically significant, $t(269) = 1.98$, $p = 0.0483$, Cohen's $d = 0.24$. However, the evidence for the group difference was rather indecisive according to the Bayes factor, which yielded equally likely support for the null and alternative models, $BF_{01} = 1.2$. Thus, we found that the participants reported a decrease in their desire for antibiotics as a consequence of experiencing the COVID-19 pandemic but tended to do more so after watching the apocalyptic post-antibiotic era film.

### The adverse effect of the intervention on adherence to prescribed antibiotics
In contrast to our expectation, the intervention did slightly decrease adherence to the prescribed course of antibiotics: 93.5% said they would take prescribed antibiotics compared with 98.4% in the baseline group, $\chi^2(1) = 4.71$, $p = 0.0299$, $\varphi = 0.13$. This difference was statistically significant, but also small (3/192 vs. 12/186 participants who did not adhere). A Bayes factor analysis yielded relative evidence supporting the null and alternative effect models equally, $BF_{01} = 1.0$. Also, the adherence did not differ between

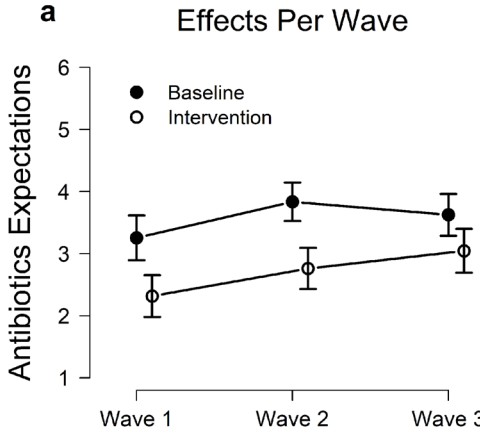

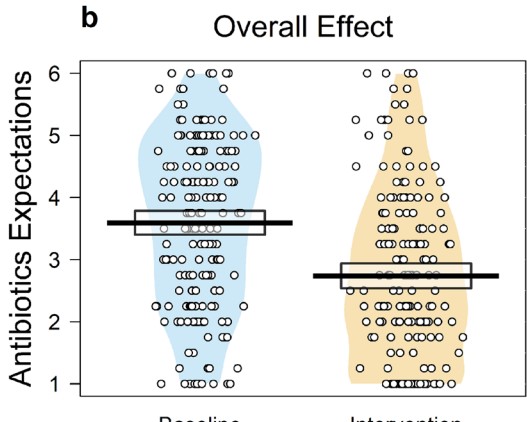

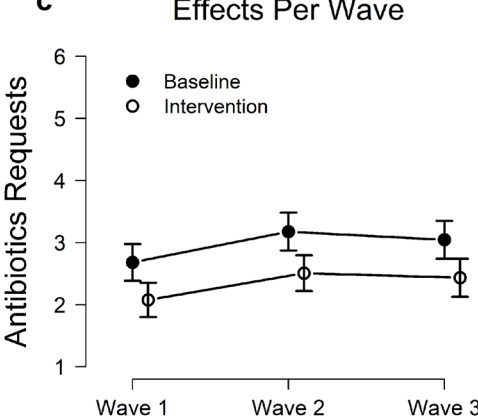

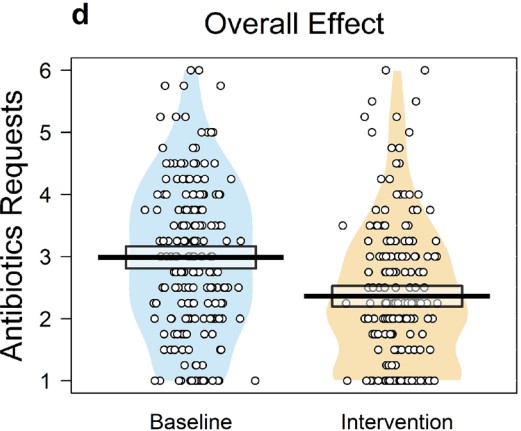

**Fig. 1 | The effects per wave and the overall effect of the post-antibiotic era film depiction on antibiotics expectations and intended antibiotics requests** . Wave 1 was the Pre-COVID wave, from 13/2/2018 to 16/03/2018, before the pandemic. Wave 2 was the COVID lockdown wave from 24/2/2021 to 28/5/2021, during and shortly after the third and last lockdown which ended on 29 March in the UK. Wave 3 was the COVID post-lockdown wave from 11/11/2021 to 02/5/2022, when restrictions were gradually lifted despite the spread of the Omicron variant. Bar graphs (**a**, **c**) depict the intervention effect on antibiotic expectations (**a**) and intended requests (**c**) in each wave. Circles represent estimated sample proportions; error bars represent 95% Agresti-Coull add-4 confidence intervals for a binomial proportion. Pirateplots (**b**, **d**) depict the overall intervention effect on antibiotic expectations (**b**) and intended requests (**d**). Horizontal bold lines represent means, boxes represent 95% confidence intervals, shaded/coloured areas (i.e., beans) represent smoothed densities and circles represent individual responses. The sample sizes for both variables were as follows—the baseline condition: overall $n = 192$, $n$ per wave: $n_1 = 55$, $n_2 = 68$, $n_3 = 69$; the intervention condition: overall $n = 186$; $n$ per wave: $n_1 = 52$, $n_2 = 68$, $n_3 = 66$.

the conditions in the individual waves, yielding more support for the null effect models: wave 1, $\chi^2(1) < 0.01$, $p = 1$, $\varphi = 0.01$, $BF_{01} = 10.5$; wave 2, $\chi^2(1) = 1.36$, $p = 0.243$, $\varphi = 0.15$, $BF_{01} = 4.2$; and wave 3: $\chi^2(1) = 3.56$, $p = 0.059$, $\varphi = 0.19$, $BF_{10} = 1.2$. Thus, only weak evidence exists about the adverse effect of such an intervention on adherence to a course of antibiotics.

## Discussion

The cinematic depiction of a post-antibiotic apocalyptic era proved to be an effective method to lower participants' expectations for clinically inappropriate antibiotics and their intentions to request them from their family physicians. The effects were robust and substantive. We also found out that our participants' experiences with the COVID-19 pandemic lowered their self-reported desire for antibiotics, but only when they were exposed to the antibiotic pandemic. Finally, we found evidence of the intervention lowering adherence to prescribed antibiotics; however, this effect was small, and supporting evidence was weak.

Our findings are aligned with the findings from previous studies testing the effect of threat-based messages on antibiotics expectations and intentions to request antibiotics[18,19]. These findings are also consistent with the effectiveness of threat-based messaging in other health-related contexts, such as disease prevention, sexually transmitted diseases, and quitting smoking and drinking[20]. Our findings are also aligned with the models

assuming the effectivity of high-level fear-based messaging such as the linear model of fear-based messaging[22]. If the apocalyptic messaging had backfired we would not have observed such robust effects on expectations and requests, but because we used a between-subjects design we cannot exclude the possibility that such messaging did backfire for some individuals. Nevertheless, apocalyptic messaging should be compared experimentally with lower levels of threat. Threat-based communication might be a powerful tool for raising awareness about antimicrobial resistance, however, with the ever-changing landscape of global health threats, it is important to recognise the limitations of relying continuously and solely on this type of messaging. Threat-based messaging might be counterproductive if the imminent threat is already recognised; continued exposure to such messaging could potentially result in the effect plateauing. In addition, it is important to acknowledge that threat-based messaging might work on average, but other messages might induce larger changes in different groups of the public. Therefore, research on public health messaging about antimicrobial resistance should test a multimodal and contextualised approach to messaging whenever it is feasible.

Several limitations of the presented study are noteworthy and should be addressed in future research. First, since the film Catch represents a complex intervention, future research should try to disentangle the possible synergistic effects of the embedded emotional story triggering empathy,

raising awareness about antimicrobial resistance, and showing the dramatic consequences of a post-antibiotic era. Previous research showed that emotional stories and narrative evidence can be effective in influencing risk judgments and health-related decision-making such as the uptake of vaccines[32,33]. The effect of patient stories describing the consequences of antimicrobial resistance on patients' health decreased antibiotic expectations and intentions to request antibiotics[18].

Second, while depicting an apocalyptic post-antibiotic era seems to be effective, we did not compare its effectiveness with other health messaging strategies. For instance, messages outlining negative consequences but imposing a lower level of threat or communication of the benefits of preserving modern medicine as we know it may be more or equally effective as post-apocalyptic messaging without any adverse effects on adherence[13,34–37]. Depicting the consequences of antimicrobial resistance in low-income countries occurring right now might be more effective than depicting these consequences in the future.

Third, the potential role of experience with the COVID-19 pandemic in the increased plausibility of post-antibiotic pandemics, as assumed in our hypothesis, should be approached with caution. This experience was not subject to experimental manipulation, it was complex and evolved over time. Alternative explanations could account for our findings. For instance, the growing levels of vaccine hesitancy and declining trust in health authorities during the later stages of the COVID-19 pandemic might have affected trust in antibiotic recommendations[38]. This seems unlikely in our case, as we would also observe a reduction in adherence to antibiotics as advised by the family physician across the waves. This was not the case. Nevertheless, other explanations for the self-reported decline in the desire for antibiotics warrant future investigation.

Finally, we did measure one obvious adverse effect of such health messaging by focusing on adherence to judiciously prescribed antibiotics, but other adverse effects should be considered as well. For instance, the post-antibiotic future can trigger negative emotions such as anxiety and some message avoidance behaviours. Recent findings showed that to motivate social distancing during COVID-19, messages focusing on potential losses triggered higher levels of negative emotions than those focusing on potential gains, despite their similar effectiveness[39]. Possible adverse effects ought to be investigated in future research.

To conclude, the depiction of a post-antibiotic apocalyptic era using the short film Catch effectively lowered the expectations for clinically inappropriate antibiotics and intentions to request antibiotics for a hypothetical illness. The intervention also interacted with the participants' experience of the COVID-19 pandemic and lowered their self-reported desire for antibiotics. Possible adverse effects on lowered adherence to treatment courses should be proactively managed in public health messaging.

## Data availability
The data set[28] used in this research, including the specific numerical data underlying Fig. 1 (source data), is available at Open Science Framework: https://osf.io/3fvqm/, https://doi.org/10.17605/OSF.IO/3FVQM.

## Code availability
The R code[29] used in this research is available at Open Science Framework: https://osf.io/bfek5/, https://doi.org/10.17605/OSF.IO/BFEK5.

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

## Acknowledgements
We sincerely thank Ellie J. Dadds, Holly S. L. Dingli, and Ibili Fidan for their help with the data collection in Wave 1.

## Author contributions
MS: Conceptualisation, Methodology, Software, Formal Analysis, Investigation, Resources, Data Curation, Writing – Original Draft, Writing – Review & Editing, Visualisation, Project Administration; MJ: Resources, Writing – Review & Editing

## Competing interests
The authors declare no competing interests.
