## [Peer Review File · Communications Medicine]

Reviewers' comments:

Reviewer #1 (Remarks to the Author):

The manuscript "Seeing an Apocalyptic Post-Antibiotic Future Lowers Antibiotics Expectations and Request" reports on a repeated cross-sectional intervention study investigating the effects of different films on antibiotic expectations. It is shown that a post-antibiotic film reduces antibiotic expectations more strongly than a control film.

In my view, this research addresses an important question. I appreciate the application of open science standards, including preregistration and publicly sharing data and analyses. I also like the idea to evaluate real-world intervention materials in a controlled setting (including a control condition). That said, there are some issues that reduce my enthusiasm for this manuscript because they hinder the conclusions drawn from the study. Some of these issues might be addressed by further analyses, others might require a discussion of potential limitations. I detail my major concerns below.

1. There are several problems because different samples have been recruited for each of the waves. In this design, it is difficult to attribute differences in responding to time differences and associated situational effects (e.g., the COVID-19 pandemic). To address the problem of potential sample heterogeneity, I suggest adding demographic controls to the main analyses as an exploratory analysis. There are also other statistical options to potentially address this concern, such as propensity score matching.
2. Related to the point above, it is not clear to me why and in what direction the COVID-19 pandemic should affect antibiotic expectations. As far as I understand, the authors argue that the pandemic experience might increase the plausibility of a post-antibiotic scenario. However, there are several other effects that could play a role here. For instance, the increasing levels of vaccine hesitancy and decreasing trust in health officials during (the later stage of) the COVID-19 pandemic might have affected trust in antibiotics and associated antibiotic recommendations as well. I appreciate that the authors are careful in their interpretations of time effects (which may be due to sampling differences, as suggested above, or various motivations induced through the COVID-19 pandemic). Nevertheless, given that this is a major limitation of the current study, I would suggest discussing such potential effects a bit broader.
3. The videos may differ in various aspects. This is mentioned in the discussion and is a natural limitation when using realistic intervention materials. Still, I think the manuscript would benefit from a somewhat more detailed discussion of such differences and their potential effects on the results. For instance, it could well be that the intervention video simply increases knowledge about antimicrobial resistance and its potential negative consequences, irrespective of the fear appeal. In other words, there may be an informational and an affective value of the intervention video, both of which might have an impact on the intervention effect.

Reviewer #2 (Remarks to the Author):

Thank you for the opportunity to review this interesting study.

It is refreshing to see an empirical study which tests communication and messaging resources slash interventions for the public to address antimicrobial resistance.

The rationale, methods, results and discussion are clearly articulated.

The methods section has sufficient detail and is replicable.

There are no major issues which require addressing.

The study could add valuable knowledge to help shape world antibiotic Awareness Week as well as year round campaigns/interventions. It will also help understand the utility of fear based framing for other public health issues.

A particular strength of the study is the three time points of data collection crossing COVID.

I have a few observations which the authors may wish to consider if/when revising the manuscript.

It would be helpful to pick up in the discussion the potential limitations of fear based public Health communication when more apparently imminent public health threats emerge. Or how the method can flex.

The authors rightly refer to fear based messaging for other public health concerns but a little bit more discussion about that learning may help the readers situate the current study (e.g. alcohol misuse smoking cessation and climate change).

While higher levels of fear may lead to behavior change the evidence does seem to suggest a plateauing of the effect - "Resistance" emerging to messaging on resistance.

So in discussion it would help to have an explicit recognition that multimodal approaches are needed as different strategies will work for different population groups.

Stronger mention that some of the scenarios that appear to be a future scenario are actually evident and current in many low middle income countries affecting the most vulnerable already.

Minor suggested amends:

Page 7 line 121: "fear based messaging can be effective..." - does this refer to AMR or more generally?

Page 7 line 123 - 2126 - has the theory been empirically tested who is it a suggestion?

Suggest replacing the word 'appeals' as it appears on page 7 line 129 and in other places with 'campaigns' or 'interventions'.

Thank you again.

Revision letter

Reviewers' comments:

Reviewer #1 (Remarks to the Author):

The manuscript "Seeing an Apocalyptic Post-Antibiotic Future Lowers Antibiotics Expectations and Request" reports on a repeated cross-sectional intervention study investigating the effects of different films on antibiotic expectations. It is shown that a post-antibiotic film reduces antibiotic expectations more strongly than a control film.

In my view, this research addresses an important question. I appreciate the application of open science standards, including preregistration and publicly sharing data and analyses. I also like the idea to evaluate real-world intervention materials in a controlled setting (including a control condition). That said, there are some issues that reduce my enthusiasm for this manuscript because they hinder the conclusions drawn from the study. Some of these issues might be addressed by further analyses, others might require a discussion of potential limitations. I detail my major concerns below.

RESPONSE: Thank you for your positive assessment and constructive criticism of the manuscript. We agree with the raised issues. We have followed your suggestions to re-analyse the data while controlling for sociodemographic variables to strengthen our conclusions. We have also discussed more thoroughly the limitations in the Discussion section.

1. There are several problems because different samples have been recruited for each of the waves. In this design, it is difficult to attribute differences in responding to time differences and associated situational effects (e.g., the COVID-19 pandemic). To address the problem of potential sample heterogeneity, I suggest adding demographic controls to the main analyses as an exploratory analysis. There are also other statistical options to potentially address this concern, such as propensity score matching.

RESPONSE: This is a very good point. Thank you for the suggestion. We have added demographic variables as control variables into the main analysis as an exploratory analysis. In brief, the main effect of the intervention remained unaffected in terms of statistical significance; the same was true for the differences observed between data collection waves. We reported the analysis in full in the manuscript and listed it as an additional exploratory analysis.

2. Related to the point above, it is not clear to me why and in what direction the COVID-19 pandemic should affect antibiotic expectations. As far as I understand, the authors argue that the pandemic experience might increase the plausibility of a post-antibiotic scenario. However, there are several other effects that could play a role here. For instance, the increasing levels of vaccine hesitancy and decreasing trust in health officials during (the later stage of) the COVID-19 pandemic might have affected trust in antibiotics and associated antibiotic recommendations as well. I appreciate that the authors are careful in their interpretations of time effects (which may be due to sampling differences, as suggested above, or various motivations

induced through the COVID-19 pandemic). Nevertheless, given that this is a major limitation of the current study, I would suggest discussing such potential effects a bit broader.

RESPONSE: Thank you for providing this astute observation which led us to a more exhaustive interpretation of the findings. We extended our discussion of the possible role of the COVID-19 pandemic experience in antibiotic expectations to other possible explanations.

Our original assumption was indeed that the pandemic experience might increase the plausibility of a post-antibiotic scenario. This was because some opponents of the apocalyptic terminology argue that it is not realistic, yet, the experience of the COVID-19 pandemic made this a more realistic possibility. It is true, however, that experiencing the pandemic was more complex and changing over time. For instance, decreasing trust in public health in the later stage of the pandemic might have affected the trust in antibiotic recommendations. Decreased trust in prescribers, rather than increased plausibility of post-antibiotic pandemics, could account for the self-reported decrease in the desire for antibiotics. However, if this were the case, we would observe a decrease over time in adherence to antibiotics as recommended by the family physician. This was not the case. We found no significant change across the three different waves in adherence. Nevertheless, there may be other explanations for the self-reported decline in the desire for antibiotics that warrant future investigation.

3. The videos may differ in various aspects. This is mentioned in the discussion and is a natural limitation when using realistic intervention materials. Still, I think the manuscript would benefit from a somewhat more detailed discussion of such differences and their potential effects on the results. For instance, it could well be that the intervention video simply increases knowledge about antimicrobial resistance and its potential negative consequences, irrespective of the fear appeal. In other words, there may be an informational and an affective value of the intervention video, both of which might have an impact on the intervention effect.

RESPONSE: We appreciate your suggestion. As is the case with any complex intervention, isolating the impact of a single dimension is a challenge, a fact that is now more clearly acknowledged and discussed in the Discussion section. Specifically, we discuss the possible role of raising awareness of antimicrobial resistance, providing an engaging story, and activating emotions such as empathy. Despite the interplay of the intervention's components, we believe that the post-antibiotic apocalypse picture was one of the main driving forces of the effect as indicated by the participant's responses to the open-ended question concerning the aspect of the film that has affected their attitudes towards antibiotics.

Reviewer #2 (Remarks to the Author):

Thank you for the opportunity to review this interesting study.

It is refreshing to see an empirical study which tests communication and messaging resources slash interventions for the public to address antimicrobial resistance.

The rationale, methods, results and discussion are clearly articulated.
The methods section has sufficient detail and is replicable.

There are no major issues which require addressing.

RESPONSE: Thank you for your positive assessment and constructive criticism of the manuscript.

The study could add valuable knowledge to help shape world antibiotic Awareness Week as well as year round campaigns/interventions. It will also help understand the utility of fear based framing for other public health issues.

A particular strength of the study is the three time points of data collection crossing COVID.

I have a few observations which the authors may wish to consider if/when revising the manuscript.

It would be helpful to pick up in the discussion the potential limitations of fear based public Health communication when more apparently imminent public health threats emerge. Or how the method can flex.

RESPONSE: This is a very valid point. We now discuss in more detail the potential limitations of threat-based messaging amidst more apparently imminent public health threats. Currently, threat-based communication might be useful for raising awareness about the looming threat of antimicrobial resistance. However, once antimicrobial resistance becomes a more apparent threat, the fear-messaging might become counter-productive yielding strong anxiety or paralysing action. Thus, flexibility is vital in public health communication strategies. While fear-threat messaging is valuable for developing the currently lacking awareness about antibiotic resistance, a multifaceted approach that includes empathy and empowerment might be more effective once the threat is more apparent.

The authors rightly refer to fear based messaging for other public health concerns but a little bit more discussion about that learning may help the readers situate the current study (e.g. alcohol misuse smoking cessation and climate change).

RESPONSE: Thank you for the suggestion; this might be useful for the readers. We have added the context of such studies in the Introduction.

While higher levels of fear may lead to behavior change the evidence does seem to suggest a plateauing of the effect - "Resistance" emerging to messaging on resistance.

RESPONSE: This is a very valid point. We now discuss in more detail the potential limitations of such messaging in the Discussion section.

So in discussion it would help to have an explicit recognition that multimodal approaches are needed as different strategies will work for different population groups.

RESPONSE: This is an astute observation. We fully agree with the usefulness of such a multimodal approach. We are currently working on such projects. The multimodal approach is now acknowledged in the Discussion.

Stronger mention that some of the scenarios that appear to be a future scenario are actually evident and current in many low middle income countries affecting the most vulnerable already.

RESPONSE: This is sadly true: some of the scenarios described in the introduction are already happening in some countries. It would be worthwhile for future research to test the impact of such messaging described as a possible future or as a current situation in a different country and compare its effectiveness. We consider this in more detail in the Discussion.

Minor suggested amends:

Page 7 line 121: "fear based messaging can be effective..." - does this refer to AMR or more generally?

Page 7 line 123 - 2126 - has the theory been empirically tested who is it a suggestion?

Suggest replacing the word 'appeals' as it appears on page 7 line 129 and in other places with 'campaigns' or 'interventions'.

Thank you again.

RESPONSE: Thank you for the suggestions. We found them all useful and amended the text accordingly.

REVIEWERS' COMMENTS:

Reviewer #1 (Remarks to the Author):

I thank the authors for their responsive revision. In my view, the revision adequately deals with the few limitations (e.g., by mentioning and discussing them) despite its many strengths. I have no further comments or concerns and look forward to see this paper published.

Reviewer #2 (Remarks to the Author):

Thank you for carefully taking into consideration the suggested revisions.

No further suggestions.